# Home-Delivered Meals: Characterization of Food Intake in Elderly Beneficiaries

**DOI:** 10.3390/nu13062064

**Published:** 2021-06-16

**Authors:** Ségolène Fleury, Virginie Van Wymelbeke-Delannoy, Bruno Lesourd, Paul Tronchon, Isabelle Maître, Claire Sulmont-Rossé

**Affiliations:** 1Centre des Sciences du Goût et de l’Alimentation, AgroSup Dijon, CNRS, INRAE, Université de Bourgogne Franche-Comté, F-21000 Dijon, France; virginie.vanwymelbeke@chu-dijon.fr (V.V.W.-D.); claire.sulmont-rosse@inrae.fr (C.S.-R.); 2Saveurs et Vie, 94390 Orly, France; paul.tronchon@saveursetvie.fr; 3CHU Dijon Bourgogne, F-21000 Dijon, France; 4Département de Gériatrie CHU, 63000 Clermont-Ferrand, France; lesourd.bruno@orange.fr; 5USC 1422 GRAPPE, Ecole Supérieure d’Agricultures (ESA), SFR 4207 QUASAV, INRAE, 55 rue Rabelais, 49007 Angers, France; i.maitre@groupe-esa.com

**Keywords:** older adults, nutritional intake, home care services, meals-on-wheels, protein-energy malnutrition

## Abstract

*Objective.* In this study, we focus on elderly people (≥70 years old) benefiting from a home delivery meal service as part of a social welfare program. We aimed to: (i) assess the gap between the recommended and actual nutritional intake in this population and (ii) study the relationship between the intake of nutrients and the variables characterizing the participants’ health and nutritional status. *Design.* A dietary survey (24-hour record) was conducted during a home interview, with 64 people receiving a home delivery meal service (75% women; 70–97 years old). At the same time, the participants answered questionnaires assessing their nutritional and health status. *Results.* Our data showed that the consumption of 70 to 80% participants was not sufficient for reaching the nutritional recommendations for energy and macronutrients. Additionally, the data showed that the lower the energy and protein intakes, the higher the risk of malnutrition. In addition, one third of the participants were both overweight or obese and at risk of undernutrition or undernourished. Our study demonstrated that the heavier the person, the more difficult it was for them to meet the nutritional recommendations based on kilograms of body weight. Finally, individuals receiving two to three delivered meals per day had higher energy and protein intakes than those receiving a single meal. *Conclusion.* These results suggest that it is important that home meal delivery companies improve the quality of their meals and service so that their recipients can better meet nutritional recommendations.

## 1. Introduction

With advancing age, the onset of physical and/or cognitive disabilities may lead older adults to seek help in activities of daily living. In the home, this assistance is essentially focused on shopping or meal preparation [1], and it can be provided by a family member (child, spouse) or a professional caregiver (household helper, life assistant). Among the various care offers, meal delivery service (also known as meals-on-wheels) can provide dependent elderly people with a solution for maintaining balanced and regular nutritional intake [2]. Several studies have shown a positive impact of home delivered meals on nutritional intake and/or nutritional status [3,4,5,6,7,8,9,10,11,12,13,14,15]. For example, Roy & Payette, (2006) showed that, among people over 65 years who requested a homecare service, the implementation of meal delivery service resulted in a significant increase in energy and protein intake after eight weeks of follow-up, while no change was observed among the people who did not subscribe to this service [11]. However, several studies have also shown that a significant proportion of beneficiaries did not meet their nutritional needs [13,16,17,18,19,20,21,22]. For example, in the study conducted by Ponza, (1996), which included 818 beneficiaries of an American home-delivered meal service, 44% of beneficiaries did not meet one-third of the energy recommendations and 14% did not meet one-third of the protein recommendations [21]. 

In Anses report, which was a response to the referral n° 2012-SA-0103 in order to update the nutritional references [23], a person’s energy needs only decrease by 7 to 18% between 40–49 years and 60–69 years. In the elderly population, the French Health Authorities (HAS, 2007) recommends a daily intake of 30 kcal per kg of body weight [24]. Regarding protein intake, work that was conducted by the PROT-AGE consortium Bauer et al., [25] and by the European Society for Enteral and Parenteral Nutrition (ESPEN) [26] recommended an intake of 1 to 1.2 g per day and per kg of body weight beyond the age of 60 (1.5 g in the case of illness), when compared to 0.8 to 1 g per kg of body weight in younger individuals. Several factors may explain this increase in protein requirements with age. On the one hand, advancing age is accompanied by changes in the digestive system that may alter nutrient absorption [27]. On the other hand, there is resistance to the positive effects of dietary protein on protein synthesis, a phenomenon that is known as anabolic resistance. For instance, it has been shown that a denser bolus of amino acids is needed for stimulating muscle protein synthesis in older people as compared to younger individuals [28,29]. Finally, the elderly may also have higher protein requirements to compensate for a higher base metabolism in inflammatory conditions. In the elderly, insufficient food intake is associated with an increased risk of undernutrition [30]. Subsequently, undernutrition is associated with numerous negative consequences in this same population, including muscle loss, impaired immune defenses, poor wound healing, aggravation of existing diseases, and impaired functional and muscular capacities [31,32]. Without care, it can induce or aggravate frailty and/or dependence, which can finally affect the quality of life and life expectancy of the elderly [33,34]. Conversely, Salminen et al. (2020) showed that adequate energy intake can prevent frailty and maintain a good quality of life [35].

In France, meals-on-wheels involves 80,000 people. This activity belongs to the medico-social sector of collective catering alongside hospital and nursing home catering [36]. This service can be offered by local authorities or associations in the social action framework. In fact, 66% of the French municipalities accounting for 5000 to 20,000 inhabitants offer a meal delivery service to the most deprived and/or dependent elderly [37]. However, the older population benefitting from a meal delivery service has been seldom studied in France, according to the systematic literature review conducted by Fleury et al. [38]. In the present study, we have characterized the food intake of the elderly people benefiting from a home-delivered meal service provided by the social services of the city of Paris. The objective of our study was to: (i) evaluate energy and macronutrient intakes and the gap between consumption and recommendations; and (ii) study the relationship between food intake, weight status, and nutritional status, as well as observe the influence of physical and health status on the previous factors.

## 2. Materials and Methods

### 2.1. Participants

People that were over 70 years old receiving at least five meals per week from the home meal delivery service of the Social Services of Paris were sent a letter describing the study, and they were then contacted by telephone. Those who agreed to participate in the study were visited by a dietician to obtain their informed consent after presenting the study and providing complete documentation. Individuals suffering from an acute illness at the time of the visit were excluded from the study. The sample size was calculated from the prevalence of people not meeting the dietary recommendations, as recently reported by Borkent et al. [39]. In this study, 71% of the population had an inadequate intake that met the recommended protein intakes. With a 95% confidence level and a margin of error set at 10%, the study sample size was set as 79 participants. We were secured with a 20% dropout rate and planned to recruit approximately 100 individuals.

### 2.2. Interviews

A dietitian visited each participant at home for a face-to-face interview lasting approximately 60–90 min. During this session, detailed data were collected through questionnaires and tests. The experimental protocol was approved by the appropriate French ethics committee (CPP ESTI 2015/24—IDRCB N° 2015-A01324-45).

### 2.3. Socio-Demographic Data

The socio-demographic variables included age, gender, marital status (couple, single, widowed), education level (none, primary, secondary, higher), and self-perceived financial situation (fragile, average, comfortable).

### 2.4. Food Intake Measurement

Food intake was assessed using a 24-hour record on a meal delivery day (for participants not receiving a meal every day—excluding weekends) [40,41,42]. Oral nutritional supplements were not prescribed to all participants whose nutritional status required them and, therefore, were not considered in the food intake measurement. The dietitian called the participant in the morning to remind him/her to carefully record their food intake for the day. The dietitian filled out a record book with the participant during the visit the following day. When the participants had difficulty remembering what they had eaten the day before, the dietitian would check the home delivery meals menu and check their refrigerators to see whether there were some leftovers. Macronutrient intakes (energy, protein, carbohydrate, fat), total daily nutrient intakes (TDIs), and daily nutrient intakes relative to body weight (DNIs) were determined from the Ciqual French Food Composition Table (2016) [43].

#### 2.4.1. Nutritional Status Characterization

The nutritional variables included Body Mass Index (BMI) measurement and the Mini-Nutritional Assessment (MNA). Body weight was measured using an electronic scale (TERRAILLON^®^). The participants were weighed with their clothing, and the weight was adjusted by subtracting the average weight of the clothes they wore [44]. Height was recorded from the person’s ID card or it was calculated based on the lower leg length using the Chumlea formula [45]. The lower leg length was measured using a height gauge, with the ankle and knee at a 90° angle. The MNA is an 18-item questionnaire that includes anthropometric measures (weight, arm, and calf circumferences), as well as nutrition and health questions. This questionnaire defines a score between 0 and 30 [46]: a score below 17 indicates undernourished status; a score between 17 and 23.5 indicates a risk of undernutrition; and a score above 23.5 indicates normal nutritional status.

#### 2.4.2. Physical, Psychological, and Cognitive Status Characterization

Comorbidities. The participants were asked to report any illnesses during the interview and, when possible, provide a copy of their latest medical prescriptions. A physician (co-author BL) analyzed these data to determine the number of comorbidities for each participant (e.g., cardiovascular disease, neuropsychiatric disease, metabolic disease…).

Functional capacities. Participants took two of the Short Physical Performance Battery (SPPB) tests [47]: chair lift, which is the time that is required to stand up from a chair without using the armrests five times in a row, and walking speed over four meters without assistance. Because many participants were unable to complete the tests without the help of armrests (36 participants out of 60) or a cane/walker (21 participants), the scores that were proposed in the SPPB were adapted to allow for a better gradation of the participants’ physical capacities (Table 1). The chair lift and walking speed scores were added together to obtain a score between 0 (worst functional performance) and 18 (best functional performance).

Cognitive status. The participants completed the Mini Mental State Examination (MMSE) questionnaire. The MMSE consists of 11 questions that assess the following cognitive abilities: orientation, learning, attention, memory, language, and praxis. The score ranges from 0 (worst cognitive performance) to 30 (best cognitive performance) [48].

Depression. Participants completed the short version of the Geriatric Depression Scale (GDS) questionnaire. This version includes 15 items [49,50]. The respondents answered “yes” or “no” for each question (e.g., “Are you satisfied with your life?”; “Do you get bored often?”). The GDS score ranges from 0 to 15 (the higher the score, the more depressed the person).

### 2.5. Data Analysis

Descriptive variables were presented as percentages or means (M) associated with standard deviations (SD). Firstly, a univariate mixed linear model was performed with energy and protein intakes as dependent variables, and each of the other variables were measured as independent variables. Secondly, multiple linear regressions were only conducted with the independent variables that are associated with a significant trend or effect (*p* < 0.10). These regressions were performed using the GLM procedure of the SAS software (SAS Institute Inc., Cary, NC, USA).

## 3. Results

### 3.1. Participant Characterization

At the time of the study, there were 1607 older people receiving home meal delivery service of the Social Services of Paris. Approximately half of these people were randomly selected to be contacted. However, after seven months of recruitment and 694 telephone calls, we managed to include only 64 participants (proportion of women: 75%; average age: 83 ± 7 years; age range: 70–97 years). One-third of those contacted did not answer the telephone and 84% of those contacted refused to participate in the study. The main reasons for refusal were fatigue or the constraints associated with seeing a dietician at home.

Table 2 displays the characteristics of the participants included in the study. In terms of the frequency of meal delivery, 44 participants received one meal per day maximum (between four and seven meals per week, mainly lunch) and 20 participants received between two and three meals per day (between 13 and 21 meals per week, including breakfast, lunch, and dinner). A low level of education was reported by 41% of the participants (‘no education’ or ‘no longer in school after primary school’) and 60% reported being in a fragile financial situation. The participants had rather low functional abilities (average BPSS score of 7.2/18). On average, the participants had three comorbidities and a high BMI, with 55% of the participants being overweight or obese.

### 3.2. Nutritional Intakes

Table 3 presents the total daily nutrient intakes (TDIs) and daily nutrient intakes relative to body weight (DNIs), which were determined for macronutrients (energy, protein, carbohydrates, and fat) from food records. These were compared to the recommended dietary allowances (RDAs) for this population (Bresson & Mariotti, 2016; Deutz et al., 2014). The results show that between 70% and 80% of the participants do not eat enough to meet the nutritional recommendations. For these participants, the average deficit between recommended and total intakes is about 872 (SD = 524) kcal, 33 (SD = 20) g protein, 105 (SD = 59) g carbohydrate, and 35 (SD = 21) g fat.

### 3.3. Nutritional Status

Table 4 presents the prevalence of nutritional risk in the total sample and by weight status categories. According to the Mini Nutritional Assessment, 39 participants (61%) were at risk of undernutrition and 11 (17%) were undernourished. It is interesting to note that, among the participants at risk of undernutrition or undernourished, 24 participants (48%) were overweight or obese. The prevalence of nutritional risk appears to be higher among obese individuals (13 out of 14; 93%) than among overweight individuals (11 out of 21; 52%). Nevertheless, this result should be confirmed in a larger population. Finally, all of the underweight individuals were at risk of malnutrition or undernourishment.

### 3.4. Univariate Regression Results

Table 5 presents the results of the linear univariate regression models for energy and protein TDIs and DNIs. No significant relationship was observed between the energy or protein intakes and socio-demographic variables (gender, age, marital status, education, and financial status). TDIs were significantly and positively associated with nutritional status, as defined by the MNA score and the number of co-morbidities. DNIs were positively associated with the SPPB score (significant for energy, trend for protein). Weight status (BMI) was significantly associated with TDIs and DNIs, except for total daily energy intake (positively with TDIs and negatively with DNIs). Finally, energy and protein intakes, both TDIs and DNIs, were all significantly and positively associated with the number of meals delivered per day.

### 3.5. Multiple Regression Results

In multiple linear regression models, the total daily energy intake remained significantly associated with the MNA score (β = 38.44; CI: [17.46/59.43]; *p* < 0.001) and the number of comorbidities (β = 70.78; CI: [20.92; 120.63]; *p* < 0.01). Similarly, the total daily protein intake was significantly associated with the MNA score (β = 2.10; CI: [0.85/3.35]; *p* < 0.001) and the number of comorbidities (β = 5.44; CI: [2.16; 8.71]; *p* < 0.01). While the total daily energy and protein intakes increased with the number of co-morbidities, we found that the lower the total daily energy and protein intakes, the greater the nutritional risk (the lower the MNA score). 

Energy intake was significantly associated with BMI when intakes are divided by the body weight (DNIs) (β = −0.56; CI: [−0.83; −0.30]; *p* < 0.001). Figure 1 illustrates the relationship between DNIs and BMI: as BMI increased, the daily intake relative to body weight decreased. Thus, the average daily energy intake per kg body weight was 21.2 kcal (SD = 6.2) for people with normal weight status and 15.1 kcal (SD = 3.8) for people with obesity. When considering the recommendation of 30 kcal/kg body weight per day, a corollary of this result is that people had greater difficulty in meeting the recommendations as BMI increased. A similar result pattern was observed for protein per kg body weight (β = −0.02; CI: [−0.04; −0.01]; *p* < 0.01), with a daily intake per kg body weight of 1.0 (SD = 0.4) for people with normal weight status and 0.7 (SD = 0.2) for obese people, while the recommendation is for 1.2 g protein per kg body weight. 

Finally, the energy and protein intakes were still associated with a significant effect of the number of meals delivered per day in a multiple linear regression model. Thus, people receiving two or three meals per day had higher energy and protein intakes than people receiving only one (total energy: β = 340.13; CI: [175.24; 505.02]; *p* < 0.001; total protein: β = 16.11; CI: [175.24; 505.02]; *p* < 0.001). Similar patterns were observed for the energy and protein intakes relative to body weight.

## 4. Discussion

### 4.1. Inadequate Dietary Intake in View of Nutritional Recommendations

An important finding in the present study is that a large proportion of elderly people receiving home-delivered meals that were provided by social services were not meeting their daily nutritional needs according to current recommendations. Specifically, 70% to 80% of the participants did not eat enough food to reach their energy and/or macronutrient (protein, carbohydrates, and lipids) needs. For these people, the energy and protein intakes were only about two-thirds of the recommended amount (respectively, 68% and 72%). These results are in line with those that were reported in the systematic literature review that we conducted on a similar population [38]. Several authors have shown that the energy and/or protein intakes of elderly people receiving home-delivered meals were, on average, lower than recommended [15,16,19,20]. Table 6 presents a short list of studies that have estimated the prevalence of individuals who do not meet their energy and/or protein requirements (studies that were extracted from the systematic literature review by Fleury et al., 2021 [38]). The prevalence of individuals who did not meet the recommended intakes was not negligible in any study, even if the calculation methods vary from one study to another, and this prevalence tended to be higher for energy than for protein [13,21,22,51].

In France, home-delivered meal services are required to comply with a regulatory guide, the GEM-RCN. This guide is an official text that was issued by the French government to provide a framework for the nutritional quality of meals that are served by medico-social catering services. This guide sets up portion sizes and nutritional content of the meals in order to cover the nutritional needs of the target population [52]. Accordingly, the home-delivered meals should account for approximately 100% of the individual’s nutritional requirements when a catering service provides an older individual with three meals per day. In our study, it was interesting to note that nutrient intake tends to increase with the number of meals delivered: people receiving two to three meals a day had higher energy and protein intakes than people receiving only one meal a day. This result should be taken with caution, because it is possible that intake measurements may have been more accurate for delivered meals than for self-prepared meals, as discussed in the study limitations. However, it is interesting to relate these results to the work of An (2015), Walden et al. (1989), and Walton et al. (2015) [3,13,14]. These three authors compared the nutritional intakes of beneficiaries on a day with meal delivery and on a day without. These three studies found that the energy and protein intakes were lower when people prepared their own meals than when they received meals-on-wheels service. In addition, Fleury et al. (2021) showed that one-third of their population receiving home-delivered meals did not consume their delivery all at once, but preferred to spread the food over several meals (e.g., the main course is eaten at lunch, while the starter, the bread, and the cheese are eaten at dinner) [2].

In our sample, 69% of the participants received no more than one meal per day. However, other reasons can be proposed to explain the insufficient nutritional intake that was observed in the present study, regardless of the number of meals that were delivered per day. On the one hand, some studies reported the dissatisfaction of older people regarding the sensory quality of home-delivered meals [2,53,54]. For instance, users’ complaints relate to meat texture (too hard), seasoning, or menu variety. On the other hand, several factors that are related to aging have been shown to have a negative impact on appetite and food intake [55]. Some are related to the physiological processes of aging (less efficient digestion, alteration of hormonal mechanisms regulating hunger and satiety, decline in the ability to perceive food odors and flavors, and loss of teeth). Other social factors can be implicated in decreased food intake and/or increased nutritional risk, for example, insufficient financial resources or loneliness (e.g., as a result of widowhood) [56,57], which has been associated with a decrease in calorie intake [58] and food variety [59]. Finally, factors that are related to physical capacities and health status may alter food intake. For example, physical disabilities affecting one’s ability to purchase groceries and/or cook may affect food choices and nutritional intake [57]. The univariate regressions in our study showed a relationship between the energy and protein intakes per body weight and the simplified SPPB score reflecting participants’ motor skills: the more mobile the participants were, the higher the protein intake per kg body weight (this effect disappeared in multiple regression). Finally, the presence of comorbidities (e.g. cancer, depression, cardiovascular disease, and chronic infection) is frequently associated with a loss of appetite and an increase in nutritional risk [60]. However, in our study, we observed a negative correlation between the total energy and protein intakes and the number of comorbidities: the more comorbidities a person has, the higher his/her nutrient intakes are. This result is difficult to explain, and future research will have to focus on clarifying this link between food intake and chronic illness, particularly when considering the type of illness and the fatigue and pain associated with each condition. 

Insufficient food intake (especially energy and protein) is accompanied by an increased risk of undernutrition, as indicated in the introduction. Here, we observed a correlation between the MNA score, which reflects nutritional status, as well as total energy and protein intakes: the lower the intakes, the higher the risk of undernutrition. A correlation between the MNA score and energy intake has already been reported [61,62]. More recently, the Global Leadership Initiative on Malnutrition (GLIM) consortium reviewed the criteria for diagnosing undernutrition, and included “decreased food intake” as an etiological diagnostic criterion for the adult population [30].

The insufficient food consumption observed in our study and reported in the scientific literature should lead home-delivery meal companies to further develop their offer. Two ways could be explored to enable the beneficiaries of a home-delivery meals service to better reach the nutritional recommendations. A good place to start could be improving meals’ quality, from both a nutritional and sensory perspective. From a nutritional point of view, the meals delivered could be enriched (i.e., increase the energy and/or protein density of the meals without increasing the portion size), which has been recognized as relevant for increasing the food intake of small eaters [63]. From a sensory point of view, the sensory qualities of the meals delivered could be optimized (meat texture, seasoning) to better match the expectations and preferences of elderly consumers. This approach, which consists of reworking recipes that are based on the results of tests carried out among the elderly, has proven to be effective in increasing eating pleasure and food intake in retirement homes [64,65]. A second way could be to promote more complete home-delivered meal services, covering more meals and not just one meal per day of a few meals per week. One of the first barriers to resolve is obviously of an economic nature. Economic constraints meant that many older people received a limited number of meals, despite social assistance. However, beyond this key issue, several older people are reluctant to receive a complete offer because they want to maintain a certain autonomy regarding their food—to keep some freedom in relation to their meals [2]. Increasing the user-friendliness of these services by improving, for example, the presentation of meals, the choice of menus and even the social environment of the meal may be an effective lever for improving the perception of home-delivered meal services and facilitating their acceptance by those who need it [57]. It would be interesting for home-delivery meals services to consider diversifying their offer (more menu options, possibility of having additional food products delivered) in order to provide their beneficiaries an opportunity to better balance food consumption throughout the day. In fact, several studies found this approach to be effective in retirement homes [65,66,67,68,69]. 

### 4.2. A Double Misfortune: Being Simultaneously Overweight and at Risk of Undernutrition 

While being underweight (BMI ≤ 18) is a criterion of undernutrition [24], being overweight or obese does not protect against undernutrition. In our study, 55% (*n* = 35) of the participants were overweight or obese. Among these participants, 69% (*n* = 24) were at risk of undernutrition or undernourishment, which represented 37% of the total sample. Data from the scientific literature show that the prevalence of nutritional risk (also measured with the MNA questionnaire) varies from 21% [70] to 50% [71] for older people with a BMI greater than 25. In overweight or obese individuals, the progressive loss of muscle mass is often accompanied by an increase and redistribution of fat mass [72], and it often goes unnoticed [73,74]. This “sarcopenic obesity” is associated with an increased risk of comorbidities and mortality: Batsis et al. (2014) showed that obese and sarcopenic women had a higher risk of mortality than non-obese or non-sarcopenic women [75].

The results of our study showed a positive and significant association between nutritional intake in relation to body weight and BMI: the higher a person’s BMI, the more the daily energy and protein intake by kg of body weight decreases. This expected result raises the question of whether it is appropriate to define nutritional recommendations according to weight. Estimating recommendations based on resting energy expenditure might be an alternative. However, the formula put forward by Harris and Benedict [76] to calculate this expenditure has not been validated for the elderly population. Another alternative would be to estimate the requirements based on an “adjusted” weight (e.g., corresponding to a BMI of 25), but the notion of an “ideal” weight is not very relevant in overweight patients. However, this result also highlights the “double misfortune” that is experienced by overweight or obese elderly people. Firstly, overweight or obese people have an increased risk of developing metabolic diseases [77] and, secondly, they have difficulty in meeting nutritional recommendations and may, therefore, have an increased risk of undernourishment. The issue of undernutrition in overweight people is likely to become a major issue in the future, because of both the aging of the population and the increasing prevalence of overweight and obesity in the general population. In France, it is estimated that 63% of people aged between 65 and 74 and 57% of people over 75 are overweight or obese [78]. However, the management of overweight elderly people at risk of undernutrition is complex (see the literature review published by Porter Starr et al., 2014 [79]). It seems that interventions combining a diet for weight loss and regular physical exercise are effective if an adequate protein intake is maintained [80].

### 4.3. Limitations of the Study

The relatively small sample size is the first limitation of the study, which limits the generalization of our results (this limitation is found in two similar studies that are mentioned in Table 6: Lipschitz et al., 1985, Walden et al., 1998 [13,51]). Our initial goal was to recruit about 100 participants. However, after seven months of recruitment and 694 telephone calls, we only managed to include 64 participants. Other authors have also described difficulties recruiting dependent elderly people, especially those benefiting from a home-delivered meal service. For instance, Houston et al. (2015) and Ziylan et al. (2017) only managed to include 14% of the elderly people that they contacted [42,81]. The researchers contacted 486 and 300 individuals, respectively, for a final recruitment of 68 and 42 participants. In the study conducted by Arjuna et al. (2018), the inclusion rate was only 24% for a final enrollment of 29 participants. In our study, it is worth noting that the participants were recruited among people benefiting from social services in Paris [40]. These participants would be expected to have relatively low incomes and education levels, and previous studies have suggested that recruiting participants with a low socio-economic status was five to six times more difficult than recruiting participants with an intermediate socio-economic status [82,83].

A second limitation of the study concerns the recording of food consumption. Although performing three 24-hour records is considered to be a reference method for assessing nutrient intakes [84,85], we chose to perform only one 24-hour record so that the constraints that are associated with the research would remain acceptable to the participants, who are easily fatigued. Arjuna et al. (2018), Gollub & Weddle (2004), and Ziylan et al. (2017) used a similar approach—these studies also involved elderly people who received home-delivery meals service [40,41,42]. In order to ensure that the measurement was as accurate as possible, the dietitian called the participant to remind him/her to record all his/her consumption and possibly to keep the leftovers on the measurement day. Finally, during the interview, the dietitian used the delivery meal menus to help people remember what they had eaten and looked at the leftovers in the refrigerator.

## 5. Conclusions

Among the elderly people benefiting from home-delivered meals that are offered by the social services of Paris, the consumption of seven to eight out of 10 participants was below the current recommended nutritional intake for energy and macronutrients. The nutrient intake tended to increase with the number of meals delivered: people receiving two to three meals a day had higher energy and protein intakes than people receiving only one meal a day. In this population, insufficient energy and protein intake was accompanied by an increased risk of undernutrition. In fact, a correlation was observed between the nutritional status, as measured by the MNA questionnaire, and the total energy and protein intakes: the lower the intakes, the higher the risk of undernutrition.

These results should be viewed with caution, as they rely on a single 24-hour dietary record. Additional research is needed to confirm them, but also to explore why home-delivered meal beneficiaries do not reach the recommended intakes. However, the present study, in line with pervious published studies, suggests that it is important for meal delivery companies to optimize the quality of their meals and services to improve their beneficiaries’ likelihood of eating well and maintaining their health.

## Figures and Tables

**Figure 1 nutrients-13-02064-f001:**
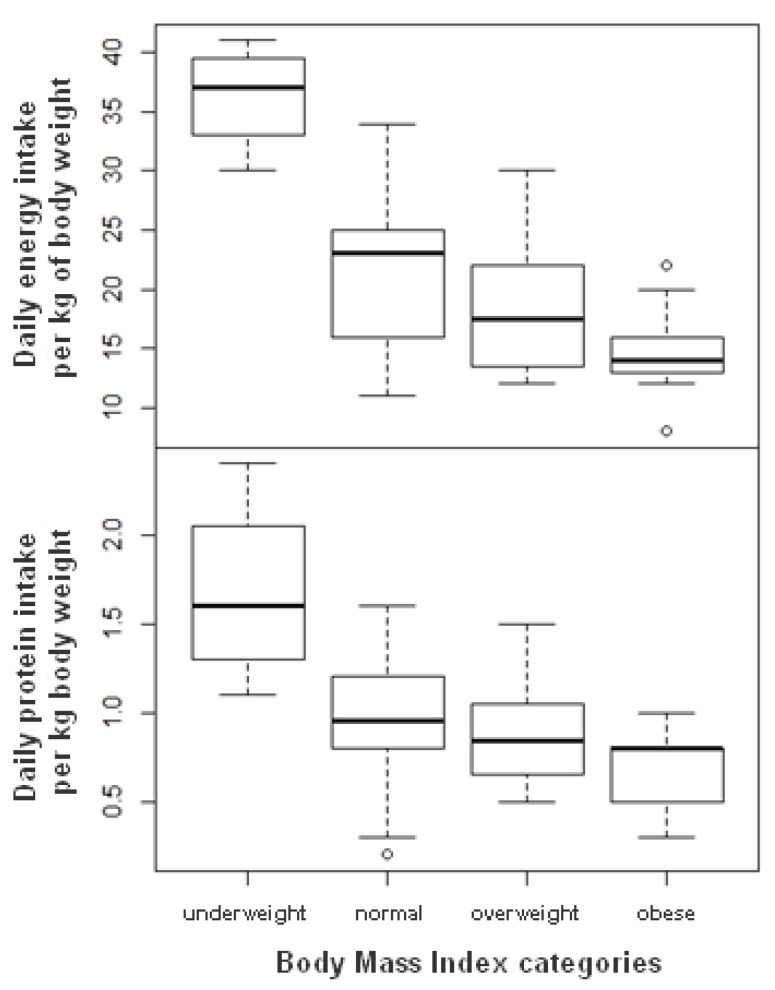
The boxplots of daily energy and protein intakes relative to body weight (DNIs) by weight status (BMI) of participants.

**Table 1 nutrients-13-02064-t001:** The adaptation of chair lift and walking speed scores (SPPB).

SPPB Initial Scores	Modified Scores
**Chair lift ^a^**	
0: test not performed 1: test duration ≥ 16.7 seconds 2: test duration [16.6, 13.7] seconds 3: test duration [13.6, 11.2] seconds 4: test duration ≤ 11.1 seconds	0: no chair lift 1: 1 or 2 chair lifts 2: 3 or 4 chair lifts 3: test duration ≥ 16.7 seconds with help 4: test duration [16.6, 13.6] seconds with help 5: test duration [13.6, 11.2] seconds with help 6: test duration ≤ 11.1 seconds with help 7: test duration ≥ 16.7 seconds without help 8: test duration [16.6, 13.6] seconds without help 9: test duration [13.6, 11.2] seconds without help 10: test duration ≤ 11.1 seconds without help
**Walking speed ^b^**	
0: test not performed 1: test duration ≥ 9.3 seconds 2: test duration [6.7; 9.2] seconds 3: test duration [5.2; 6.6] seconds 4: test duration ≤ 5.1 seconds	0: test not performed 1: test duration ≥ 9.3 seconds with help 2: test duration [6.7, 9.2] seconds with help 3: test duration [5.2, 6.6] seconds with help 4: test duration ≤ 5.1 seconds with help 5: test duration ≥ 9.3 seconds without help 6: test duration [6.7, 9.2] seconds without help 7: test duration [5.2, 6.6] seconds without help 8: test duration ≤ 5.1 seconds without help

^a^ Chair lift with help: the person used the chair armrests to get up; ^b^ Walking test with help: the person used a cane or walker to walk.

**Table 2 nutrients-13-02064-t002:** Participant characteristics. Means are presented with standard deviations (in parentheses). The values in brackets are the minimum and maximum values.

Variables	Participants (*n* = 64)
% of women	75%
Age	83.4 (7.5) (70–97)
Number of meals delivered per week	
≤7 meals per week	69%
Between 13 et 21 meals per week	31%
Marital status	
Alone ^a^	51%
Couple	44%
Widow	5%
Education level	
No	14%
Primary	27%
Secondary	33%
Graduate	26%
Self-perception of financial resources	
low	60%
Fair	31%
Good	9%
Nutritional statut	
Body Mass Index (BMI)	26.1 (6.1) (15–46)
Underweight	6%
Normal body weight	39%
Overweight	33%
Obese	22%
Mini Nutritional Assessment (MNA)	20.1 (3.8) (10–26)
Normal	22%
Risk of malnourishment	61%
Malnourishment	17%
Physical, psychological, and cognitive status	
Number of comorbidities	3.4 (1.6) [0–8]
Short Physical Performance Battery (SPPB)	7.2 (5.2) [0–14]
Chair lift	3.0 (2.4) [0–8]
Walking test	4.2 (3.4) [0–8]
Mini-Mental State Examination (MMSE)	25.9 (3.9) [14–30]
Geriatric Depression Scale (GDS)	4.9 (3.4) [0–14]

^a^ Single and divorced. The MNA score varies from 0 to 30 (the higher the score, the better the nutritional status). The BPSS score varies from 0 to 18 (the higher the score, the better the functional abilities). The MMSE score varies from 0 to 30 (the higher the score, the better the cognitive performance). The GDS score varies from 0 to 15 (the higher the score, the greater the depression).

**Table 3 nutrients-13-02064-t003:** Daily Nutrient Intakes (TDI and DNI) as compared to Recommended Nutrient Intake (RNI). Means are presented with their standard deviation (in parentheses).

Nutrients	TDI	DNI	RNI per kg of Body Weight	% Deficient ^1^
Energy (kcal)	1306 (369)	20.0 (7.0)	30.0	83%
Proteins (g)	58 (21)	0.9 (0.4)	1.2	72%
Carbohydrates (g)	152 (48)	2.4 (1.1)	3.5	81%
Lipids (g)	48 (18)	0.8 (0.3)	1.1	75%

TDI: Total Daily nutrient intake; DNI: Daily Nutrient Intake relative to body weight; RNI: Recommended Nutrient Intake; ^1^ Percentage of participants whose DNI were below the RNI per kg of body weight for the macronutrient of interest.

**Table 4 nutrients-13-02064-t004:** The prevalence of nutritional risk (MNA) by weight status categories (BMI).

	Body Mass Index (BMI)				
Nutritional Status (MNA)	Underweight <21	Normal 21–24	Overweight 25–29	Obese >30	Total
Normal > 23.5 *n* %	0 0%	3 4.5%	10 16%	1 1.5%	14 22%
At risk of malnutrition 23.5–17 *n* %	4 6%	13 20%	10 16%	12 19%	39 61%
Malnutrition < 17 *n* %	5 8%	4 6%	1 1.5%	1 1.5%	11 17%
Total *n* %	9 14%	20 31%	21 33%	14 22%	64 100%

**Table 5 nutrients-13-02064-t005:** The result of analyses in the univariate mixed linear model. The β coefficients are associated with their 5% confidence interval (CI) and the significance threshold: (*): *p* < 0.10; * *p* < 0.05; ** *p* < 0.01; *** *p* < 0.001.

	Energy				Protein			
	TDI		DNI		TDI		DNI	
	β	CI	β	CI	β	CI	β	CI
Sex (reference: man)								
Woman	−77.73	[−299.36; 143.90]	1.02	[−3.34; 5.39]	−9.37	[−22.09; 3.35]	−0.04	[−0.28; 0.19]
Age	−8.91	[−22.04; 4.21]	−0.10	[−0.36; 0.16]	−0.53	[−1.30; 0.23]	−0.01	[−0.02; 0.01]
Marital status (reference: single)								
Couple	141.92	[−310.18; 594.02]	2.75	[−6.11; 11.61]	14.36	[−11.52; 40.25]	0.31	[−0.16; 0.79]
Widow	−42.41	[−242.47; 157.64]	1.77	[−2.15; 5.69]	−5.38	[−16.83; 6.08]	0.03	[−0.18; 0.24]
Education level (reference: no)								
Primary	−113.28	[−428.94; 202.37]	−1.59	[−8.03; 4.89]	−5.74	[−24.30; 12.82]	−0.08	[−0.41; 0.26]
Secondary	142.80	[−158.81; 444.42]	−0.50	[−6.70; 5.70]	1.62	[−16.12; 19.35]	−0.08	[−0.41; 0.24]
Graduate	154.55	[−157.65; 466.76]	0.50	[−5.92; 6.91]	10.53	[−7.83; 28.89]	0.16	[−0.17; 0.50]
Self-perception of financial resources (reference: low)								
Fair	−52.80	[−263.89; 158.29]	2.29	[−1.83; 6.42]	−2.69	[−15.05; 9.67]	0.06	[−0.16; 0.29]
good	−190.44	[−582.78; 201.90]	−1.68	[−9.35; 5.98]	−6.43	[−29.40; 16.54]	0.22	[−0.19; 0.63]
BMI	12.09	[−3.34; 27.52]	−0.60 ***	[−0.87; −0.34]	1.11 **	[0.23; 1.97]	−0.02 **	[−0.04; −0.01]
Number of meals delivered per week (reference: ≤1 meal per day)								
>1 meal/day	335.76 ***	[151.39; 520.13]	5.64 **	[1.91; − 9.38]	16.50 **	[5.41; 27.59]	0.31 **	[0.11; 0.51]
MNA	27.42 *	[2.73; 52.11]	−0.09	[−0.59; 0.42]	1.48 *	[0.04; 2.93]	−0.01	[−0.04; 0.01]
Comorbidities	65.64 *	[7.13; 124.15]	−0.06	[−1.26; 1.14]	5.56 ***	[2.32; 8.79]	0.04	[−0.02; 0.10]
SPPB	−0.84	[−19.55; 17.87]	0.43 **	[0.08; 0.78]	−0.17	[−1.26; 0.92]	0.02 (*)	[0.00; 0.04]
MMSE	3.20	[−22.60; 29.00]	−0.04	[−0.55; 0.47]	−0.09	[−1.60; 1.41]	−0.01	[−0.03; 0.03]
GDS	1.12	[−27.77; 30.02]	−0.07	[−0.64; 0.49]	0.84	[−0.83; 2.51]	0.01	[−0.02; 0.04]

TDI: Total Daily nutrient Intake; DNI: Daily Nutrient Intake relative to body weight; BMI: Body Mass Index; MNA: Mini-Nutritional Assessment; SPPB: Short Physical Performance Battery; MMSE: Mini-Mental State Examination; GDS: Geriatric Depression Scale.

**Table 6 nutrients-13-02064-t006:** The percentage of elderly people benefiting from home-delivery meals service and who do not meet their energy and protein needs.

Author (s), Year Country	Population	Energy Intakes	Protein Intakes
Our Study France	64 beneficiaries of a home-delivery meals service (social services center of the city of Paris)	83% did not meet the recommendations of 30 kcal per day per kg of body weight	72% did not meet the recommendations of 1.2 g of protein per day per kg of body weight
Lipschitz et al, 1985 USA	33 beneficiaries of a home-delivery meals service (not OAA)	35% did not reach 80% of the energy and protein recommendations *.	
Ponza, 1996 USA	818 beneficiaries of a home-delivery meals service (OAA)	44% did not reach 2/3 of the energy recommendations (1900–2300 kcal/d)	14% did not reach 2/3 of the protein recommendations (50–63 g/d)
Sharkey, 2003 USA	279 beneficiaries of a home-delivery meals service (OAA)	25% did not reach 2/3 of the energy recommendations *	25% did not meet the protein recommendations *
Walden et al, 1998 USA	16 beneficiaries of a home-delivery meals service (OAA)	56% did not meet the energy recommendations *	6% did not meet the protein recommendations *

OAA: Older American Act. * The authors did not specify in their article which recommendations they relied on to determine prevalence.

## Data Availability

To request the complete data tables please contact the corresponding author.

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
