# Peer review of "Home-Delivered Meals: Characterization of Food Intake in Elderly Beneficiaries"

_nutrients, 2021, doi:10.3390/nu13062064_

Round 1

Reviewer 1 Report

The manuscript presents an interesting study and very relevant considering the importance nowadays of the food intake of the elderly people benefiting from a home-deliveredmeal service provided by the social services. Elderly people are of great importance for sustainable prevention policy.

Besides,  the authors highlighted and described clear the difficulties recruiting dependent elderly people, especially those benefiting from a home-delivered meal service compared to nursing houses or elderly care houses. Indeed, elderly is a special needs group.

Author Response

Dear Editor and dear Reviewers,

We thank you for your interest in our research and for your advices. We have revised our paper according to your suggestions and we do hope that we succeed in answering your questions. Anyway, we remain at your disposal for any further information.

We thank the reviewer for his encouraging comment. We have added some context information related to the French meals-on-wheels situation in the introduction to provide more background information in the introduction (lines 81-88). The conclusion has been also rephrased to better fit with the results (Lines 444-458)

Best regards,

Ségolène Fleury

Reviewer 2 Report

This study looks at intakes and nutritional status of elderly people receiving home delivered meals. The study found that people were not meeting their recommended intakes but people receiving 2-3 meals were better-off than those receiving only a single meal per day.  The authors concluded that the nutritional quality of meals and the service needed to be improved to help maintain adequate nutrition.

Table 3 needs to explain TDI and DNI.

Percentages should be included in Table 4.

Abbreviations need to be defined in Table 5.

The adequate nutrition of elderly people is clearly and important topic but this study is very small and does not provide any information on how sample size was determined. Also need to provide information on the number of people invited to get 64 who agreed before the limitations.

The method for the dietary data collection is not clearly described. It is called a 24-hr recall but participants were reminded to record their food intake for the day so not a recall.

There is no information on why people do not reach the recommended intakes which limit the conclusions and suggestions for improvement.  Is the aim of the meal delivery service to provide 100% of requirements or is the expectation that there are other sources of food? Are people not consuming the food because they don’t like it, because they are not hungry or something else?

The observation that people receiving more delivered meals consume more energy suggests the meals do help, would be important to know what is limiting the people who have fewer.

A single 24 hr record/recall is not appropriate for this study.

Author Response

Dear Editor and dear Reviewers,

We thank you for your interest in our research and for your advices. We have revised our paper according to your suggestions and we do hope that we succeed in answering your questions. Anyway, we remain at your disposal for any further information.

Please find below the answers to your questions,

Best regards,

Ségolène Fleury

Table 3 needs to explain TDI and DNI.

Done.

Percentages should be included in Table 4.

Done.

Abbreviations need to be defined in Table 5.

Done.

The adequate nutrition of elderly people is clearly and important topic, but this study is very small and does not provide any information on how sample size was determined. Also need to provide information on the number of people invited to get 64 who agreed before the limitations. A paragraph providing information on sample size determination has been added (lines 101-106). Information about recruitment have been moved from the ‘Limitation’ paragraph (discussion) to the beginning of the result section (‘Participants characterization’ section) (lines 179-185).

The method for the dietary data collection is not clearly described. It is called a 24-hr recall but participants were reminded to record their food intake for the day so not a recall. The term 24h-recall has been changed into 24h-record in the manuscript.

There is no information on why people do not reach the recommended intakes which limit the conclusions and suggestions for improvement.  Is the aim of the meal delivery service to provide 100% of requirements or is the expectation that there are other sources of food? Are people not consuming the food because they don’t like it, because they are not hungry or something else? The discussion was reviewed to answer your different comments. Regarding the nutritional “objective” of meal delivery services in France: please see lines 288-298. Regarding possible reasons why our participants did not meet the nutritional recommendations: please see lines 309-316.

The observation that people receiving more delivered meals consume more energy suggests the meals do help, would be important to know what is limiting the people who have fewer. This point is now discussed on lines 364-373.

A single 24 hr record/recall is not appropriate for this study. We agree with the reviewer that a single 24-h record is a limit of the present study, as discussed on lines 432-442. We have added this limit in the conclusion (lines 453-454). In fact, in a recent talk, Jonh Mathers raised the issue of measuring dietary intake (Mathers, 2021. Advances in nutrition and health research over the past 10 years and a glimpse into the future. 6th JPI concerence, 20-21 April 2021). Mathers elaborated on the need for better tools and technologies to achieve reliable dietary intake. This is even true for dependent older people, who are reluctant to be involved in research protocol because of fatigue and decline capacities. In the present study, we had to find a compromise between scientific constraints and older people’s capacities, to prevent dropouts among our sample – that was by the way already very difficult to recruit. Consequently, a choice was made to restrict dietary record to a single day, in line with previous authors that have faced the same difficulties as us. Another option would have been to ask an experimenter to call the participants after each meal, but we feared that this strongly biases the data by providing much more social interaction around the meal than in everyday life. We hope that the use of new technologies will enable better dietary intake measurements in the older population (for instance, by taking pictures of their meals). However, these technologies should ensure to fit with older people capacities, especially dependent older people who often aged over 80 years old, a generation that is not comfortable with the use of new technologies.

Reviewer 3 Report

The manuscript entitled “Home-delivered meals: characterization of food intake in elderly beneficiaries” by Fleury and co-authors describes nutritional intake in elderly population that becomes their meals through home delivery meal services. The study also evaluated correlations between intake of nutrients, nutritional status and BMI.

This study is of high quality, data is carefully analysed and results are well presented. Study limitations, including low number of participates, are discussed, conclusions and discussion are easy to follow. The difficulties that scientist have when recruiting participants are good described and, actually, are similar to our experience. Additionally, presentation of existing literature is very good and important for this kind of studies.

Author Response

Dear Editor and dear Reviewers,

We thank you for your interest in our research and for your advices. We have revised our paper according to your suggestions and we do hope that we succeed in answering your questions. Anyway, we remain at your disposal for any further information.

We thank the reviewer for his encouraging comment. It is reassuring to know that the difficulties we are encountering in our research work are shared by other research teams.

Best regards,

Ségolène Fleury

Round 2

Reviewer 2 Report

The authors have appropriately responded to my comments.